# Venous Thromboembolism Recurrence in Latvian Population: Single University Hospital Data

**DOI:** 10.3390/medicina55090510

**Published:** 2019-08-21

**Authors:** Valdis Ģībietis, Dana Kigitoviča, Sintija Strautmane, Kitija Meilande, Verners Roberts Kalējs, Anastasija Zaičenko, Kristīne Maķe, Aivars Lejnieks, Andris Skride

**Affiliations:** 1Department of Internal Diseases, Riga Stradiņš University, 4 Hipokrāta iela, LV-1079 Riga, Latvia; 2Pauls Stradiņš Clinical University Hospital, 13 Pilsoņu iela, LV-1002 Riga, Latvia; 3Faculty of Medicine, Riga Stradiņš University, 16 Dzirciema iela, LV-1007 Riga, Latvia; 4Riga East Clinical University Hospital, 2 Hipokrāta iela, LV-1038 Riga, Latvia

**Keywords:** venous thromboembolism, pulmonary embolism, recurrence, cancer, anticoagulation

## Abstract

*Background and objectives*: Recurrence of venous thromboembolism (VTE) after a primary event is common; however, no sufficient risk scores have been widely introduced in clinical practice. The aim of this study was to assess the risk factors for VTE recurrences, as well as the effect of treatment strategies on the recurrence rate in a single-center patient cohort. *Materials and Methods*: The prospective cohort study included consecutive patients in a single center from June 2014 till June 2018 presenting with acute VTE confirmed by imaging tests. All patients were followed up for at least one year or till death. Statistical analyses were conducted using IBM SPSS Statistics 23 and Stata 13. Competing risk of death was considered. *Results*: A total of 219 eligible patients were identified during the study period. Pulmonary embolism with or without deep vein thrombosis (DVT) was present in 95.9% (*n* = 210), isolated DVT was present in 4.1% (*n* = 9) of patients. The total number of documented recurrences was 13 (5.9%). Incidence rate was 5.6 per 100 person-years. Recurrent VTE predicted significantly higher mortality rate (hazard ratio (HR) 6.64 [95% CI 2.61–16.93]). In univariate analysis, active cancer was associated with higher recurrence rate (*p* = 0.036). In competing-risks regression model (with death as the competing risk), active cancer (subdistribution hazard ratio (SHR) 2.11 (95% CI 0.58–7.76)) did not retain statistical significance for VTE recurrence. Discontinuation and duration of anticoagulant treatment (≤6 or >6 months), and drug class in acute or long-term therapy (parenteral, vitamin K antagonist (VKA), direct oral anticoagulant (DOAC)) were not associated with recurrences (*p* > 0.05). *Conclusions*: Patients who experienced recurrent VTE had 6.6-fold higher mortality rate than patients with no recurrences. The presence of active cancer was not a statistically significant risk factor for recurrence when taking into account the competing risk of death. Duration and drug class of anticoagulation did not seem to impact recurrence rate.

## 1. Introduction

Venous thromboembolism (VTE), comprised of deep vein thrombosis (DVT) and pulmonary embolism (PE), is the third most common cause of death from cardiovascular disease after acute myocardial infarction and stroke [1,2]. Recurrence of VTE after a primary event is common with an overall incidence rate of 4.9 per 100 person-years in non-cancer patients. Higher recurrence risk has been associated with an “unprovoked” primary VTE events and malignancy-related VTE with a 1-year risk of approximately 10% and 15%, respectively [3,4]. Previously described demographic and clinical risk factors for recurrence are male gender, body-mass index, ethnic background, lower socioeconomic status, thrombophilia, hormonal therapy, pregnancy, and cancer, among others. Several multivariate risk scores such as HERDOO2 score [5], Vienna prediction model [6], DASH score [7] have been developed but due to the lack of sufficient external validation, none of them have yet been included in international guidelines [8]. Recent studies have demonstrated the superiority of long-term direct oral anticoagulants (DOAC) over vitamin K antagonists (VKA) and low-molecular-weight heparins (LMWH) in preventing recurrent VTE [9]. The aim of this study was to assess the possible clinical risk factors for VTE recurrences, including comorbidities, cancer, VTE provoking factors, demographic features, laboratory data, and treatment approaches.

## 2. Materials and Methods

### 2.1. Study Population

The study cohort included inpatients admitted to cardiology and pulmonology clinics in the Pauls Stradiņš Clinical university hospital in Riga, Latvia from June 2014 till June 2018 (the hospital covers approximately 45,000 inpatients per year). Patients were eligible for our study if the diagnosis of acute VTE was confirmed by imaging studies (Doppler ultrasound, computed tomography angiography) (see Figure 1). All radiographic images were analyzed by experienced radiologists. All included patients had a recent onset of VTE symptoms during the previous 10 days, all patients were subsequently hospitalized. All patients provided written or oral consent for participation in the registry in accordance with local hospital ethics committee requirements (Ethics Committee of Pauls Stradiņš Clinical University Hospital Development Foundation, ID code: 110614-2L, date of approval: 11.06.2014.). Patients with previous VTE (history of any DVT or PE before the index event) were excluded from the present data analysis. We made all efforts to enrol consecutive inpatients admitted to the clinics. Data were recorded in a local computer-based registry.

### 2.2. Study Design

We conducted a prospective cohort study. Baseline characteristics of patients, laboratory studies, underlying conditions, and therapeutic approaches were recorded. Comorbidity burden was calculated using the Charlson Comorbidity Index. Active cancer was defined as newly diagnosed cancer, metastatic cancer (distant metastases according to the TNM classification), or cancer that is being treated. Transient risk factors were defined as surgical intervention in the past 2 months, immobility ≥ 4 days in the past 2 months, travel > 6 hours in the past 3 weeks, hormonal (estrogen, progestogen) therapy, pregnancy, or recent childbirth. Minor persistent risk factors included BMI > 30 kg/m^2^, family history of VTE, previous episode of VTE, inflammatory bowel disease, congestive heart failure, and thrombophilia. Unprovoked VTE was defined as an event with no transient or persistent risk factors, including concomitant active cancer according to the International Society on Thrombosis and Haemostasis (ISTH) definition [10]. All patients were followed up for at least one year after the index event or till death. The index time for follow-up was the date of the VTE diagnosis. Recurrence was defined as a repeated VTE event that was confirmed using imaging tests. Date of recurrence or death was registered. The obtained patient–time data was statistically analyzed.

### 2.3. Statistical Analysis

Data were expressed as absolute numbers and percentage, mean and standard deviation (SD), median and interquartile range (IQR), incidence rate (IR), hazard ratio (HR), subdistribution hazard ratio (SHR), and confidence interval (CI), where appropriate. SPSS Statistics version 23 (IBM, Chicago, Illinois, USA) and Stata version 13 (StataCorp, College Station, Texas, USA) were used for the analyses. Univariate analyses were conducted using the Mann-Whitney U test, χ^2^-test, and Fisher’s exact test, where appropriate. Since standard regression models overestimate HRs in high-mortality patient groups, e.g., cancer patients, further analyses were done taking the competing risk of death into consideration. To assess the independent effect on mortality of statistically significant variables in univariate analysis (*p* < 0.05), competing-risks regression analysis was performed and subdistribution HRs were presented. Cox regression model with time to recurrence as a time-varying covariate was used for the assessment of the impact of recurrent VTE on mortality. A *p*-value < 0.05 was considered to be statistically significant.

## 3. Results

### 3.1. Patient Cohort

Following inclusion and exclusion criteria, 219 eligible patients were identified during the study period. Pulmonary embolism with or without DVT was present in 95.9% (*n* = 210). Isolated DVT with no imaging evidence of PE was found in 4.1% (*n* = 9) of patients. Baseline clinical characteristics of the patient cohort are shown in Table 1. The total number of documented recurrences was 13 (5.9%). Incidence rate was 5.6 (95% CI 3.3–9.7) cases per 100 person-years. Of the 13 recurrences, 12 were PE with or without DVT and 1 patient had only DVT. Five of the recurrences (38.5%) were fatal (death within 10 days of recurrence and PE as the principal cause). Median time to recurrence was 130 days (IQR 29–336). Overall all-cause mortality was 32.4% (*n* = 71).

### 3.2. Clinical Factors

Clinical factors in association with episodes of VTE recurrence and death are shown in Table 2. In univariate analysis, mean age was similar among patients with or without VTE recurrence, *p* = 0.176.

Active cancer as the single variable that demonstrated a significant association with VTE recurrence (*p* < 0.05) was further analyzed in regression models. In a simple Cox regression model, HR of VTE recurrence for cancer was 4.43 (95% CI 1.17–16.8), *p* = 0.028. A competing-risks regression model (with death as a competing risk), however, did not demonstrate a statistical significance of cancer—SHR 2.11 (95% CI 0.58–7.76), *p* = 0.259. Unprovoked versus provoked VTE was also not a significant factor for VTE recurrence—SHR 0.93 (95% CI 0.27–3.21), *p* = 0.912.

Mortality according to VTE recurrence was assessed using a Cox regression with time to recurrence as a time-varying covariate. VTE recurrences were significantly associated with higher mortality—HR 6.64 (95% CI 2.61–16.93), *p* < 0.001.

### 3.3. Anticoagulation Strategies

Acute anticoagulant treatment strategy (first 10 days of treatment)—parenteral anticoagulants, VKAs or DOACs—did not influence subsequent recurrence incidence rates during follow-up, which were found to be 9.8 (*n* = 4, *p* = 0.251) for parenteral, 5.7 (*n* = 3, *p* = 0.941) for VKAs, 4.4 (*n* = 6, *p* = 0.347) per 100 person-years for DOACs. The principal drug class (LMWH, VKA, DOAC) used during long-term treatment (over 10 days after index VTE event) was also not associated with the recurrence rate—0 (*p* = 0.738), 7.4 (*n* = 4, *p* = 0.463), 4.9 (*n* = 8, *p* = 0.591) per 100 person-years, respectively.

## 4. Discussion

Our study demonstrated a significantly higher mortality among patients who experienced recurrent VTE during a follow-up period, with an approximately 6- to 7-fold higher mortality rate. Similar results were recently reported from the COMMAND VTE (COntemporary ManageMent AND outcomes in patients with Venous ThromboEmbolism) Registry [11] with an adjusted HR of 3.24 in comparison with the 6.64 in our study. These results emphasize the importance of prevention of recurrent VTE for patient survival.

Using a simple Cox regression model, active cancer seemed to be a significant predictor of VTE recurrence with a 4-fold higher risk of recurrence. Previous studies have demonstrated a significant impact of cancer on VTE recurrence [4]; however, the recurrent VTE incidence rate of 23.5 cases per 100 person-years in our cohort is an overestimation, due to the high mortality in cancer patients. When taking into account the competing risk of death, the subdistribution HR was reduced to 2.11 and the association was not statistically significant. This finding along with previous reports, e.g., by Ay et al. [12], emphasizes the importance of competing-risk analysis when assessing patient cohorts with high mortality, like that of cancer patients 

In our cohort, the overall VTE recurrence rate was 5.6 cases per 100 person-years. The rate in non-cancer patients was 4.6 per 100 person-years. Martinez et al. recently reported a rate of 4.9 per 100 person-years with a peak at 11.1 per 100 person-years in the first six months in a large cohort study of non-cancer patients in United Kingdom [3]. In our results, median time to recurrence was 130 days (IQR: 29–336), thus, most of the recurrent events occurred during the first few months, after the index event. Recurrence rates in our study and most practice-based cohort studies are significantly higher than those documented in the DOAC clinical trials (2 to 3%) [13,14], possibly due to exclusion of some high-risk patient groups, a higher compliance with treatment, and better monitoring in these trials.

In contrast with previous reports, our data did not demonstrate any significant differences in the recurrence rates between the primary provoked and unprovoked VTE. This finding might be related to the characteristics of our patient cohort that largely included elderly, co-morbid patients at a relatively high baseline risk of VTE, precipitated or not by a transient provoking factor, hence, the presence of such a factor might not significantly reduce future recurrence in this group of patients. However, even analysis of patients with transient provoking factors and no persistent major or minor risk factors in comparison with the remaining individuals did not reveal lower recurrence rates, although this group of patients were considered to be at a lower risk [10].

In our study, discontinuation of anticoagulation, during follow-up, did not demonstrate a significant association with the recurrence rate. The analysis of the treatment duration (above or below 6 months) was limited by the small number of recurrences (*n* = 6) in patients who survived more than 6 months, since only these patients were included in order to be eligible for this analysis. Recurrence rate in these patients might have been associated with other intrinsic factors.

Principal anticoagulant drug class (parenteral, VKA, DOAC) used in acute or long-term treatment was not associated with the recurrence rate in our patient cohort. In randomized controlled trials performed during the last decade, DOACs have been demonstrated to be equally effective but a safer alternative to VKAs—a finding confirmed in a meta-analysis by Gomez-Outes et al. [15]. In our study, a possible confounding factor might be the socioeconomic status of patients, since VKAs are significantly more affordable than DOACs. In Latvia, the healthcare system is a universal health coverage, largely tax funded with co-payment from patients. However, during the time of the present study neither oral nor parenteral anticoagulants in outpatient setting were state reimbursed for the diagnosis of VTE. However, even in cancer patients, the most prevalent long-term anticoagulant class was DOACs.

### Limitations

Our study was limited by the relatively modest sample size, as only 13 cases of recurrent VTE were documented in our cohort. The study population was mainly composed of elderly individuals, thereby, different demographic groups were not equally represented, thus introducing a selection bias. Another source of bias was the fact that most of the patients were followed up by telephone in person or, if not possible, by proxy or a general practitioner, hence, some cases of recurrent VTE might be undocumented because of insufficient reporting or lack of follow-up. No consistent documentation of repeated laboratory studies or tests for residual thrombosis were performed, so these potential risk factors could not be assessed in the study.

Despite limitations, our practice-based cohort study gives important insights into the risk factors for recurrent VTE events. This registry is ongoing and will produce further data as the cohort size increases. The lack of sufficient tools for prediction of VTE recurrences for directed treatment emphasize the need for further studies in this field.

## 5. Conclusions

In our patient cohort, VTE recurrence rate was 5.6 cases per 100 person-years. Patients who experienced recurrent VTE had 6.6-fold higher mortality rate than patients with no recurrences. The presence of active cancer was not a statistically significant risk factor for recurrence when taking into account the competing risk of death. Duration and drug class of anticoagulation did not seem to significantly impact recurrence rate.

## Figures and Tables

**Figure 1 medicina-55-00510-f001:**
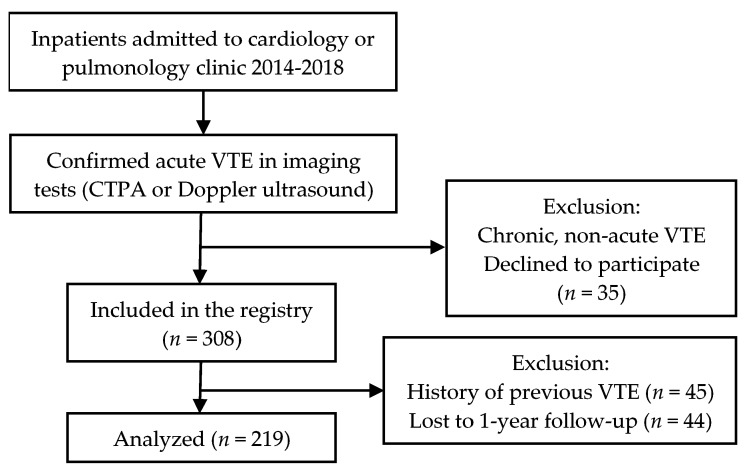
Inclusion process flowchart.

**Table 1 medicina-55-00510-t001:** Patient characteristics.

Variable	Patients (*n* = 219)
Age, years, mean ± SD	68 ± 16 (range: 19–93)
Sex, female, *n* (%)	139 (63.5)
Body-mass index, kg/m^2^, mean ± SD	28.7 ± 6.0
Obesity (BMI ≥ 30), *n* (%)	78 (35.6)
Unprovoked VTE, *n* (%)	38 (17.4)
Active cancer, *n* (%)	29 (13.2)

SD, standard deviation, BMI, body-mass index, VTE, venous thromboembolism.

**Table 2 medicina-55-00510-t002:** Univariate analysis of clinical factors in association with VTE recurrence.

Factor		Recurrences, *n* IR per 100 Person-Years (95% CI)	*p*-Value
Sex	Male	5 5.3 (2.6–10.6)	0.761
	Female	8 6.3 (2.6–15.0)	
Active cancer	Yes	3 23.5 (7.6–72.8)	0.036 *
	No	10 4.6 (2.5–8.5)	
Transient risk factors for primary VTE	Yes	4 4.8 (1.8–12.7)	0.710
	No	9 6.1 (3.2–11.7)	
Transient risk factors for primary VTE (with no persistent risk factors)	Yes	2 7.0 (1.8–28.0)	0.698
	No	11 5.4 (3.0–9.8)	
Minor persistent risk factors (excluding cancer patients)	Yes	6 4.4 (2.0–9.7)	0.833
	No	4 5.0 (1.9–13.2)	
Unprovoked VTE vs Provoked VTE	Yes	3 6.5 (2.1–20.3)	0.743
	No	10 5.4 (2.9–10.0)	
Charlson Comorbidity Index ≥4 points	Yes	10 8.3 (4.5–15.4)	0.081
	No	3 2.7 (0.9–8.4)	
Arterial hypertension	Yes	6 4.2 (1.9–9.4)	0.245
	No	7 8.1 (3.8–16.9)	
Atrial fibrillation	Yes	4 10.9 (4.1–29.1)	0.179
	No	9 4.6 (2.4–8.9)	
Anemia	Yes	4 6.5 (2.4–17.2)	0.728
	No	9 5.3 (2.8–10.2)	
Thrombocytopenia	Yes	3 10.3 (3.3–32.1)	0.287
	No	10 4.9 (2.7–9.2)	
Thrombocytosis	Yes	1 9.7 (1.4–69.0)	0.559
	No	12 5.4 (3.1–9.6)	
Obesity	Yes	5 5.2 (2.2–12.5)	0.829
	No	8 6.0 (3.0–12.0)	
Discontinuation of anticoagulation	Yes	4 3.3 (1.2–8.8)	0.129
	No	9 8.2 (4.3–15.8)	
Anticoagulation >6 months †	Yes	5 3.1 (1.3–7.5)	0.653
	No	1 1.6 (0.2–12.3)	

* *p*-value < 0.05, † Only patients who survived more than 6 months were includedIR, incidence rate, CI, confidence interval, VTE, venous thromboembolism.

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
