# Peer review of "Venous Thromboembolism Recurrence in Latvian Population: Single University Hospital Data"

_medicina, 2019, doi:10.3390/medicina55090510_

Round 1

Reviewer 1 Report

Point 1: The authors report on results of a hospital-based cohort study enrolling consecutively admitted patients with venous thrombosis. The patient cohort is a mixed elderly cohort not representing patients for example within the female fertile period. Therefore referral or selection bias must be ruled out or discussed as a potential limitation of the study.

Point 2: Among the cohort of 238 cases 17 suvjects developed recurrent thrombosis during the follow up: From my point of view only cases with a first thrombotic onset should be enrolled or adjustment for previous vascular occlusions (arterial od venous) should be performed.

Point 3: the statstical multivariable analysis should include one predictor per 10 recurrences, i.e. a maximum of n=2. However, since only active cancer is borderline significant in unvariable analysis I suggest to recalculate the data and to adjust for provoked versus unprovoked thrombosis, perhapse add the comparison blood-group 0 versus blood group non-0.

Point 4: A patient flow chart with inclusion and exlusion criteria would be helpful

Reviewer 2 Report

Comments for authors of the manuscript: Venous thromboembolism recurrence in Latvian population: single university hospital data

The paper describes a highly important complication to venous thromboembolism (VTE); recurrent VTE in a single center population. The authors have done an exhausting job in obtaining medical history, clinical and laboratory plus treatment data of nearly 300 patients. The data quality must hence be sound, and would despite the limited number of events (VTE recurrences) at the present moment contribute markedly to the available knowledge.

Overall, the data appears suitable for time-to-event analysis (i.e. calculation of incidence rates, cumulative incidences and construction of Cox proportional hazards regression models). The paper seems to aim at covering too much (line 47-48) – both assessment of risk factors for recurrent VTE and the impact of treatment of outcome. Only the first is mentioned in the headline, though. In my opinion, the study needs epidemiological revision.

I would suggest that the authors consider if the study should be an etiological or a prognostic study, see for instance Ay et al:  https://doi.org/10.1111/jth.12825. In my opinion, the latter would be problematic because of the limited number of events. I recommend the authors to describe the recurrent VTE events in this (etiological) study and save the observations concerning treatment and mortality for another paper. I have therefore not addressed the sections concerning treatment and mortality in the following. It would however be nice to depict the mortality according to VTE recurrence, preferably as a time-varying exposure.

Specific comments for the paper:

L43: The authors mention several model for assessment of VTE recurrence risk, please ad references.

L44-45: Please ad references.

L47: Could the authors please clarify which risk factors for recurrent VTE they aim at investigating – is it the ones mentioned in line 40-41? If so, please elaborate them and ad relevant references. Consider to leave out last part of L47 and 48.

Study population: please describe the setting further; how is health care provided in Latvia – is it tax-funded, how much of the  medicine expences are reimbursed (you mention in the discussion that socioeconomic sattus might confound our results due to the price of DOACS) and so on. Are patients included if they are diagnosed with VTE in the outpatient clinic or emergency departments without subsequent hospitalization?  State inclusion and exclusion criterias clearly and provide a diagram for all invited subjects, how many declined participation, how many did not have VTE diagnosed after routine examination, how many were lost to follow-up (you mentioned them in line 180) and so on -the reader must end up with a clear view of the study population.

L51: Which single university hospital? How many patients/inhabitants does this hospital cover?

L58: Which system was used, RedCap or a local setup? Was the data submission and analysis logged in that case?

60: Data were procpectively collected, and data are by nature always retrospectively analyzed; it’s not nessesary to mention this. The important part is to explain that you have person-time data at hand. Please clarify the exact “index” time (index time is mostly used in matched cohort studies for controls – you have a cohort study with prospectively collected data for a certain purpose). From which date did you follow your study participants – the VTE diagnosis date? If you followed them from onset of symptoms (10 days of symptoms you have introduced immortal time bias in your study.

62: Good point with metastatic cancer, but what if they were cured for their metastatic testis cancer? And how metastatic – does it concern both regional spread and distant metastasis  -and which classification do you use?

63: It would be great if ITSH’s definition of provoked/unprovoked VTE were applied in this study, please see https://doi.org/10.1111/jth.13336

68: The exclusion of patients who died is a methodological issue; these subjects might have died bearing an excess risk of VTE recurrence. It is never a good idea to condition on the future in time to event analysis (https://doi.org/10.1002/sim.4385) Instead the authors ought to estimate the cumulative incidence of recurrent VTE in the presence of the competing risk of death. Please see for instance the aforementioned reference by Ay et al.

Table 1. Provide solely baseline characteristics, the last four rows are not baseline information.

Table 2. A lot of information, consider to add all the internal medical information in a Charlson CI. As mentioned, the results could be presented as depicted cumulative incidences of recurrent VTE according to, for instance; active cancer versus no cancer/unprovoked VTE versus provoked VTE/ etc. In the not

Table 3. Please provide both crude and adjusted hazard ratios (and place the p-values in the last column, if it should at all be reported…) If a regression model is build, please state very clearly in the methods section which variables you decide to include, and how they are included (age as for instance – was it included in the model as dichotomous/categorical/continuous/age x age2/spline variable or other type of variable?) And select carefully what you want to adjust for – you do not have many events and the regression models thus becomes unreliable if the model includes too many parameters. With 17 events, I think you would have statistical power to include only two variables in your model.

L139-143: In this part of the discussion, I think the nature of a backward variable selection shows. If focused on p-values in a “fishing expedition” study design, biologically implausible associations may occur as it is in the nature of .05 statistics that  some associations will show by simply by chance.  

L144: ..recurrence rate of 6.2 cases per 100,000 person-years?

Round 2

Reviewer 1 Report

the manuscript has somewhat improved, however, the authors should consider the following suggestions in detail: Point 1: revise the title into... an "elderly" cohort of Lativian population.. Point 2: Line 97: "childbirth" an "oral contraceptives" should be omitted, since the age range does not cover fertile women Point 3: Patients with death not related to recurrent VTE should be censored - this has to be clearly pointed out in the ms Point 4: my previously point "selection bias" should be clear addressed in the limitation section Point 5: I clearly wish to see the analysis "provoked (including cancer)" versus "unprovoked"

Reviewer 2 Report

Please specify what a previous VTE was.  I gues it means "a DVT before the one (or only PE??) that occured in the study period"? I suggest to exclude subjects with any VTE prior to the VTE that lead to possible inclusion in this study. The number of excluded subjects should be stated in figure 1.
